# Frequency of Detection of *Candida auris* Colonization Outside a Highly Endemic Setting: What Is the Optimal Strategy for Screening of Carriage?

**DOI:** 10.3390/jof10010026

**Published:** 2023-12-29

**Authors:** Laura Magnasco, Malgorzata Mikulska, Chiara Sepulcri, Nadir Ullah, Daniele Roberto Giacobbe, Antonio Vena, Vincenzo Di Pilato, Edward Willison, Andrea Orsi, Giancarlo Icardi, Anna Marchese, Matteo Bassetti

**Affiliations:** 1Division of Infectious Diseases, IRCCS Ospedale Policlinico San Martino, 16132 Genova, Italy; laura.magnasco@hsanmartino.it (L.M.); danieleroberto.giacobbe@unige.it (D.R.G.); anton.vena@gmail.com (A.V.); matteo.bassetti@unige.it (M.B.); 2Division of Infectious Diseases, Department of Health Sciences (DISSAL), University of Genova, 16132 Genova, Italy; chiara.sepulcri@gmail.com (C.S.); nadir.ullah@edu.unige.it (N.U.); andrea.orsi@unige.it (A.O.); icardi@unige.it (G.I.); 3Department of Surgical Sciences and Integrated Diagnostics (DISC), University of Genoa, 16132 Genoa, Italy; vincenzo.dipilato@unige.it (V.D.P.); anna.marchese@unige.it (A.M.); 4Microbiology Unit, San Martino Policlinico Hospital, IRCCS for Oncology and Neurosciences, 16132 Genoa, Italy; 5Hygiene Unit, San Martino Policlinico Hospital, IRCCS for Oncology and Neurosciences, 16132 Genoa, Italy

**Keywords:** intensive care unit, ICU, infection control, prevention, horizontal transmission, healthcare policies

## Abstract

*Candida auris* outbreaks are increasingly frequent worldwide. In our 1000-bed hospital, an endemic transmission of *C. auris* was established in two of five intensive care units (ICUs). Aims of our study were to describe the occurrence of new cases of *C. auris* colonization and infection outside the endemic ICUs, in order to add evidence for future policies on screening in patients discharged as negative from an endemic setting, as well as to propose a new algorithm for screening of such high-risk patients. From 26 March 2021 to 26 January 2023, among 392 patients who were diagnosed as colonized or infected with *C. auris* in our hospital, 84 (21.4%) received the first diagnosis of colonization or infection outside the endemic ICUs. A total of 68 patients out of 84 (81.0%) had a history of prior admission to the endemic ICUs. All were screened and tested negative during their ICU stay with a median time from last screening to discharge of 3 days. In 57/68 (83.8%) of patients, *C. auris* was detected through screening performed after ICU discharge, and 90% had *C. auris* colonization detected within 9 days from ICU discharge. In 13 cases (13/57 screened, 22.8%), the first post-ICU discharge screening was negative. In those not screened, candidemia was the most frequent event of the first *C. auris* detection (6/11 patients not screened). In settings where the transmission of *C. auris* is limited to certain wards, we suggest screening both at discharge from the endemic ward(s) even in case of a recent negative result, and at least twice after admission to nonendemic settings.

## 1. Introduction

*Candida auris* is an emerging global threat, currently spreading worldwide with the number of colonized or infected patients constantly rising [1]. Recently, multiple nosocomial outbreaks of *C. auris* have been reported during the novel coronavirus-19 (COVID-19) pandemic [2,3,4,5,6,7,8], underlying the challenges in containing its spread, in particular when infection control measures are facing multiple challenges, such as during the COVID-19 pandemic. In Italy, *C. auris* has been reported in three confining regions, with most cases registered in our Liguria region, where the first *C. auris* case was reported in 2019 [9]. In our hospital in Genoa, the transmission of *C. auris* has been limited to two of five intensive care units (ICUs), in which *C. auris* remained endemic even after the COVID-19 pandemic, despite the efforts taken regarding infection control [2]. Possible reasons might be the persistence of *C. auris* in undetected environmental reservoirs and the constant presence of *C. auris*-colonized patients, both contributing to continuous transmission of *C. auris* to new patients. This particularly limited distribution of *C. auris* cases in a specific area within the institution highlights how, together with environmental disinfection measures [10] and strict contact precaution for colonized patients, early identification of new cases is pivotal for effective prevention and control strategies to tackle *C. auris* transmission within healthcare facilities [11]. Particularly in those institutions where the transmission of *C. auris* is limited in space, infection control efforts should be focused on patients discharged from endemic areas as noncolonized to detect colonization appearing later during hospitalization in order to prevent secondary outbreaks. To date, however, no solid evidence-based recommendations are available on the procedures for an effective screening and correct timing for screening for *C. auris* carriage in such cases. Indeed, healthcare systems are advised to develop screening policies after local risk assessment [12]. International experts suggested to discontinue preemptive contact isolation precautions for high-risk contact patients when three screening samples for *C. auris* taken 24–48 h apart result negative [11,12], with conditional recommendation to carry on repeated weekly screening for the whole duration of hospitalization even after this first negative results [13]. However, these recommendations are based on expert opinion and not on data from high-quality studies. A recently published pilot study from the US [14] proposed a screening protocol at admission for nursing homes and hospital ICUs consisting of two skin swabs (composite axilla, groin and nares bilaterally) tested with molecular assay for early detection of *C. auris.* The results of this study suggested that even brief contacts with nursing homes represented a substantial risk factor for *C. auris* colonization. However, no indication was provided on how to manage at-risk patients who tested negative at the first screening.

The main aim of our study was to describe the occurrence of new *C. auris* colonization and infection events outside the endemic wards, in order to add evidence for future policies on screening for colonization among patients discharged as negative from an endemic setting into other healthcare units. The secondary aim was to propose an algorithm for testing of such patients outside the highly endemic setting.

## 2. Materials and Methods

### 2.1. Study Design and Setting

This was a retrospective, monocenter study performed in a tertiary care hospital in Genoa, Italy, between 26 March 2021 and 26 January 2023.

During the 22-month study period, our hospital was served by 5 different ICUs: (i) one dedicated to cardiovascular surgical patients; (ii) one dedicated to surgical and solid organ transplant patients; (iii) one in the emergency department (also caring for COVID-19 patients when needed for epidemiological reasons); (iv) one dedicated to COVID-19 and respiratory patients (dismantled after a decrease in severe COVID-19 cases) and (v) the largest one for general and neurosurgical critically ill patients, which was in part converted to care for COVID-19 patients during the peak of cases in our region.

Since February 2020, an outbreak of *C. auris* has occurred. Despite continuous efforts to control the epidemic, an endemic *C. auris* transmission was established in the largest general ICU and the respiratory and COVID-19-dedicated ICU, with regular detection of incident cases of colonization and infection over time.

### 2.2. Patients’ Inclusion Criteria

Inclusion criteria for the present study were as follows: (i) adult age and (ii) first positive results of *C. auris* culture (at any site) or first positive molecular skin swab for *C. auris* performed outside the endemic ICUs.

### 2.3. Definitions and Description of Screening Protocol

Patients were considered to have had prior contact with the endemic ICUs if they had been admitted to the aforementioned ICUs for at least 48 h.

Colonization with *C. auris* was defined as growth of *C. auris* from at least one nonsterile site (urine, skin and/or respiratory tract) or molecular detection of *C. auris* in a composite skin swab (bilateral axilla and groin areas), in the absence of clinical signs or symptoms of infection. Multisite colonization was defined as an isolation of *C. auris* from more than one nonsterile site. Candidemia was defined as growth of *C. auris* from at least one blood culture. Blood cultures were collected upon clinical suspicion of infection according to standards of care and in the presence of symptoms suggestive of systemic infection (mainly fever in the case of candidemia and fever with local signs of inflammation in the case of septic arthritis).

Weekly urine cultures and, in mechanically ventilated patients, respiratory sample cultures were performed as a standard of care during ICU stay.

As of 26 March 2021, a local infection control protocol was implemented to rapidly detect emergent colonization and prevent further horizontal spread between patients. It was shared with all the clinical units in our hospital, although actively reinforced only in the ICU setting. Composite skin swabs of bilateral axilla and groin regions were performed for molecular testing to investigate *C. auris* carriage. Patients were screened at different timepoints during their ICU stay: at the moment of admission to the ICU, then once weekly throughout their ICU stay until diagnosed as colonized. Patients who tested negative for *C. auris* throughout their ICU stay were also screened at three additional different timepoints: at the moment of ICU discharge, at the moment of admission to a new nonendemic ward and within 48 h from admission to that ward, in order to detect possible late-occurring colonization. Contact isolation precautions were recommended in the destination, nonendemic ward until screening results came back negative.

### 2.4. Microbiological Analysis

Surveillance (bilateral axilla and groin) skin swabs (eSwab^®^, Copan Italia s.p.a, Brescia, Italy, Brescia skin swab) were collected and subjected to molecular analysis with real-time PCR to assess the presence of *C. auris* DNA. Briefly, total DNA purification from the swabs’ transport medium (i.e., Liquid Amies) was performed on a MagCore Plus II extractor using MagCore Genomic DNA Whole Blood Kit (RBC Bioscience Corp., New Taipei City, Taiwan), employing heat-based lysis, proteinase K and guanidine hydrochloride for cell lysis and protein degradation, according to the manufacturer’s instructions.

Real-Time PCR was carried out on a CFX96 Real-Time PCR System (Bio-Rad Laboratories, Hercules, CA, USA) using the RealCycler^®^ CANDIDA AURIS PCR KIT CE/IVD (declared sensitivity: 1 copy/µL; Progenie Molecular SLU, Valencia, Spain), according to the manufacturer’s instructions.

Screening samples sent for conventional culture were seeded onto Chromatic Candida, a chromogenic selective medium (Liofilchem, Roseto degli Abruzzi, Italy) and Sabouraud Dextrose Agar plates (Liofilchem). *C. auris* was identified in clinical and screening specimens with matrix-assisted laser desorption ionization-time of flight mass spectrometry (MALDI-TOF MS-Vitek MS; bioMérieux, Marcy-l’Etoile, France) using the VITEK MS software v4.0.

Antifungal susceptibility testing was carried out on strains causing invasive infection according to the Clinical and Laboratory Standards Institute microdilution method using the Sensititre YeastOne panel (Thermo Scientific, Waltham, MA, USA); minimum inhibitory concentration (MIC) values were determined for azoles, echinocandins and amphotericin B. Since no species-specific susceptibility breakpoints were available for *C. auris*, antifungal susceptibility testing results were interpreted according to the tentative breakpoints proposed by the US Centers for Disease Control and Prevention [15].

### 2.5. Ethical Considerations

This study was performed in accordance with the Helsinki declaration. This study protocol has been approved by the Ligurian Ethics Committee (registry number 308/2022). Patient consent was waived due to the retrospective nature of this study.

## 3. Results

During the study period, overall we observed 392 incident cases of colonization or infection with *C. auris* in our hospital. For epidemiological purposes, all new events of either colonization or infection were grouped together. In case the same patient experienced both events (colonization and infection), the date of the first event was considered.

Among these new *C. auris* colonization or infection cases, 84 (21.4%) were detected outside the endemic ICUs.

Overall, the detection of all new cases of *C. auris* colonization or infection, including cases detected in and outside the endemic ICUs, was stable during the study period. The epidemiological curve of new cases of *C. auris* detection is depicted in Figure 1. The median number of new cases of *C. auris* infection or colonization detected in the endemic ICUs was 42 for each trimester of the study period, while a median of 5 new cases of *C. auris* colonization or infection per trimester was detected outside the endemic ICUs. Only single cases (range 0–5) were detected in patients who did not have previous contact with the endemic ICUs.

Demographic and microbiological characteristics of included patients are summarized in Table 1. Skin colonization was the most frequent case of new *C. auris* detection (76.2%), with respiratory or urinary colonization accounting for 7.1% each, while 9.5% of patients presented *C. auris* candidemia as the first manifestation of *C. auris* detection.

Among the 84 included patients, 68 (81.0%) had a history of prior admission to the endemic ICUs and 16 (19.0%) did not. Figure 2 graphically summarizes the epidemiological characteristics of patients included in this study.

### 3.1. MICs Distribution of Tested Isolates

MIC values were available only for isolates causing candidemia (*n* = 11). All isolates were resistant to fluconazole and MIC to amphotericin B was 1 mg/L in three cases and 2 mg/L (i.e., breakpoint for resistance) in eight cases. All the isolates tested were susceptible to anidulafungin and caspofungin (anidulafungin MIC range 0.06–0.5 mg/L and caspofungin MIC range 0.06–0.25 mg/L); while one isolate had an MIC value for micafungin of 4 mg/L (i.e., breakpoint for resistance), with micafungin MIC range for all isolates 0.06–4 mg/L), while maintaining full susceptibility to other echinocandins; treatment of candidemia with caspofungin resulted in a clinical cure also in this latter case.

### 3.2. Patients without Prior Contact with the Endemic ICUs

Sixteen patients (19.0%) had no history of prior admission to any endemic ICU before detection of *C. auris* colonization or infection, and therefore had not been previously screened for skin carriage. Fifteen patients (93.8%) were already admitted to the hospital for a median of 18 days (IQR 11–25 days) before the first detection of colonization or infection; only one patient was found colonized during a day-hospital admission.

Nine patients (56.3%) tested positive at screening performed after potential nosocomial contact. Three cases of new colonization were detected in the same internal medicine ward within the same week, suggesting probable horizontal transmission, even though the patient known to be colonized with *C. auris* was transferred from an endemic ICU and had contact isolation precautions in place since the transfer. One patient was identified as colonized during his stay in a surgical department where at least one known *C. auris*-colonized patient and another patient discharged from an endemic ICU, although not screened promptly for *C. auris* colonization and not put under contact isolation precaution, were admitted at the same time. Three additional patients were found to be colonized during their stay in a high-intensity ward, where postoperative patients, frequently admitted to one of the endemic ICUs until clinically stable, are usually transferred. Although not mandatory per protocol, occasional screening with skin swab for *C. auris* carriage was frequently performed in that ward, due to the aforementioned epidemiological reasons. For the two remaining patients, no potential indirect contact with the endemic settings was identified.

Candidemia was the first manifestation in one patient (6.3%, see paragraph below). Another episode of candidemia developed after known skin colonization in a patient who exhibited the skin colonization while admitted to the ICU of another hospital of our city, where no regular transmission of *C. auris* is reported.

Demographic and microbiological characteristics of patients without prior contact with endemic ICUs are summarized in Table 1.

### 3.3. Patients with Prior Contact with Endemic ICUs

All 68 patients with a history of prior admission to the endemic ICUs had been previously screened for skin carriage during their ICU stay. The median time from the last negative screening to ICU discharge was 3 days (IQR 1–5 days) and to first detection of *C. auris* infection or colonization was 6 days (IQR 4–12 days). While for 28 of the patients there was no recent screening performed, for 40 (58.8%) of the patients the previous negative screening had been performed as recently as within the previous 7 days (26/40 had been tested with molecular tests and 14/40 with a conventional culture).

Median time to the first detection of colonization or infection was 3 days after ICU discharge (IQR 1–8 days), with 90% of the cases of new colonization or infection detected within 9 days from ICU discharge.

Fifty-seven patients (57/68, 83.8%) were screened at least once after discharge from the endemic ICU. The first screening with composite skin swab after discharge from the endemic ICU was performed a median of 1 day after discharge (IQR 1–3 days). This first screening resulted in the detection of *C. auris* colonization in 44 patients (44/57, 77.2%), while it was negative in the remaining 13 patients (22.8%). Of these latter 13 patients who tested negative at the first screening, median time from this negative screening and the first detection of colonization was 4 days (IQR 1–6 days), while the median time from ICU discharge to colonization was 5 days (IQR 2–8 days). Of note, none of these 13 patients had been screened more than once between the first negative result for *C. auris* and the first detection of colonization.

Among patients who tested positive at the first screening, 15/44 (34.1%) were tested with the molecular assay; while only 2/13 (15.4%) of those with the first negative screening were tested with the molecular test.

Of note, 11 patients (11/68, 16.2%) were never screened after discharge from the endemic ICUs and subsequently tested positive in a median of 22 days (IQR 16–35 days) after endemic ICUs discharge. Among these patients, candidemia was the most frequent first manifestation leading to the first detection of *C. auris* in six patients (54.5%).

Demographic and microbiological characteristics of patients with prior contact with endemic ICUs are summarized in Table 1.

### 3.4. Patients with Candidemia

Overall, 11 of 84 included patients (13.1%) developed *C. auris* candidemia. In three patients, candidemia occurred after known colonization (cutaneous in two cases; respiratory and urinary in one case). Time to development of candidemia after the detection of colonization was 19, 23 and 31 days, respectively. Among them, one patient developed a recurrence of candidemia after >30 days from the first episode.

Candidemia was the first manifestation of *C. auris* colonization in eight patients. Among these, seven (87.5%) had been screened for skin *C. auris* colonization but tested negative. The median time from the last negative *C. auris* screening to development of candidemia was 23 days (IQR 20–31 days). Only one patient with candidemia had not been previously screened for skin colonization because of a lack of contact with endemic ICUs in her past history, but she was cared for in a high intensity ward, where *C. auris* colonized patients were frequently admitted. Of note, screening skin samples for *C. auris* from that patient resulted in repeatedly negative results even after the onset of candidemia. Among the seven patients formerly screened, six (85.7%) were screened only during their ICU stay, while screening at the time of admission to a nonendemic ward was performed only in one case (with conventional culture testing) and resulted negative.

All patients who developed a *C. auris* invasive infection were treated with echinocandins and source control (e.g., removal of central venous catheter and surgical drainage of purulent material) whenever applicable, according to current guidelines on the treatment and management of invasive candidiasis [16]. Follow up blood cultures were available for 8/11 patients with candidemia, and the median time to the clearance of blood cultures was 3 days (IQR 2–5). All-cause mortality among patients with *C. auris* invasive infection (11 candidemia episodes and 1 septic arthritis) was 66.7% (*n* = 8/12) and death occurred after a median of 12 days from the onset of invasive infection (IQR 10–48).

### 3.5. Proposed Algorithm of Screening

The recommendations we propose for the containment of *C. auris* spread in those institutions where endemicity is limited to certain wards are summarized in Table 2. In particular, for patients admitted to a *C. auris* endemic setting who repeatedly test negative for colonization, we would recommend screening at the moment of endemic ward discharge, and most importantly, at the moment of admission to a nonendemic ward. We would suggest repeating the screening after 48 h from transfer before discontinuing preemptive isolation contact precautions. Moreover, as already suggested by other experts [11,12], repeated screening over time during the hospital stay would be advisable to exclude late-occurring colonization events.

Since *C. auris* colonization, even undetected, could persist for long periods of time, our protocol also advocates for screening *C. auris*-negative patients at the moment of new hospital admission if they had been admitted for at least 24 h to the endemic ICUs in the previous year.

In our proposed algorithm of screening we advocate, if feasible, to implement the use of molecular screening tests to allow for a quick discontinuation of resource-intensive preemptive contact isolation precautions, especially in settings like ours with a high number of at-risk patients.

## 4. Discussion

The main finding of our study is the need for specific screening protocols for patients discharged from highly endemic wards into other nonendemic settings within the same institution which we summarized in Table 2. Our observations might be generalizable to other settings where *C. auris* transmission is localized to a restricted setting (e.g., one hospital but not others in the same city or region).

The high volume of patients who became colonized or infected with *C. auris* in two ICUs at our hospital gave us the opportunity to investigate the spread of *C. auris* in an endemic setting. Indeed, to date, most studies on *C. auris* have focused on epidemic outbreaks, but little is known on how to provide infection control in a persistently endemic setting, particularly when endemicity is limited to certain wards within the same hospital. In such a peculiar setting, those patients who are discharged as not colonized from an endemic ward warrant particular attention in order to prevent secondary outbreaks.

To date, available recommendations on the management of patients at high risk of *C. auris* colonization are based on expert opinions. The Australasian Society for Infectious Diseases and Public Health England both advocate that high-risk patients (such as close contacts of known *C. auris* cases or those who had been admitted for at least one night to a *C. auris* endemic ward during the previous 12 months) should be placed under preemptive isolation precautions until three consecutive screening samples, taken at least 24 h apart, result negative [11,12]. Given the long turnaround time of *C. auris* conventional cultures, such recommendations might result in as long as 7–10 days of contact isolation precautions until all results are available. However, the implementation of preemptive contact precautions for such a long time might only be feasible in the case of a few potentially colonized patients, but might be impossible to implement in the case of high numbers of at-risk patients in settings with endemic transmission and high patient mobility within different hospital wards, for instance in high-intensity and/or surgical wards.

Screening for *C. auris* can rely on either cultural or molecular methods, which offer different advantages and disadvantages. On the one hand, conventional culture methods are less expensive than molecular ones and easy to implement without additional equipment, especially considering the other yeast cultures routinely performed in clinical microbiology laboratories. On the other hand, yeast growth is slow, and updated MALDI-ToF databases are needed for a reliable identification of *C. auris* in clinical samples. Moreover, even if adequate instruments are available, *C. auris* is hardly distinguishable from other *Candida* spp. on standard agar plates and every suspicious colony should be identified to rule out the presence of *C. auris*, making the whole process extremely time consuming. In the case of abundant growth of other yeasts of similar appearance, a few colonies of *C. auris* can be missed, causing a delay in the detection of colonization and in subsequent cohorting. On the contrary, molecular methods have a higher cost but also higher sensitivity, and they can grant more reliable and definitive results in a limited timeframe, being able to detect even low burdens of fungus, such as 1–10 colony forming units [17,18,19], and they provide definitive results within a few hours, generating actionable results and leading to a timely adoption of proper infection control strategies.

In our cohort, more than 20% of new cases of *C. auris* colonization or infection were detected outside the endemic ICUs, although most of the patients had previous contacts with the endemic setting. For patients previously admitted to the endemic ICUs, almost all cases of *C. auris* colonization or infection became evident within 9 days from ICU discharge. Moreover, even recent negative screening for skin carriage of *C. auris* performed outside the endemic setting did not exclude subsequent detection of colonization. Indeed, among patients who were screened early after ICU discharge and tested negative at the first sampling, around 23% subsequently tested positive after a median of 4 days from the previous negative screening. These results suggest that, in many cases, repeated screening over a short period of time might be needed to detect or exclude late-onset colonization. In addition, our data highlight how, if patients were not screened properly, candidemia due to *C. auris*, and not colonization of other sites, was the main clinical manifestation of the first *C. auris* detection (6/11 cases, 54.5%). These findings confirmed out initial observations, which formed the bases for the implementation and maintenance of a protocol dedicated to patients discharged from the endemic settings, and highlight the need to better focus on further prevention of *C. auris* transmission through extended screening (see our proposed algorithm in Table 2).

At our center, screening for skin carriage with molecular-based tests was suggested at three delicate timepoints: at the moment of ICU discharge, at the moment of admission to a nonendemic ward and within 48 h from the admission to this new ward. We noted that the indication for screening at ICU discharge was very difficult to implement. Indeed, discharge from the ICU is often difficult to schedule, with a possible lack of adherence to this recommendation. Therefore, the results of our study suggest that more efforts should be dedicated to increasing adherence to screening upon transfer to a nonendemic ward from an endemic ICU, and failure of screening at that timepoint should be considered a major missed opportunity in early detection of colonization and effective infection control. In our hospital, failure to screen after transfer to nonendemic wards occurred in 16% of patients, of whom 55% later developed candidemia as the first manifestation of their undetected *C. auris* colonization.

Moreover, single cases of detection of late-onset colonization or infection after initial negative screening results suggest that performing screening more than once after transfer from endemic settings might be needed. In addition, the use of a more sensitive test (e.g., molecular versus conventional culture) might increase the diagnostic yield in case of low fungal burden. While the clinical importance of low burden colonization for a patient who is to be discharged from a hospital setting to the community might be limited, in the case of a transfer to long term care services its detection might be fundamental to limit the further spread of *C. auris*.

Of note, 16 cases (19%) of *C. auris* colonization or infection did not show a direct epidemiological link with the endemic ICUs. This might indicate that more cases than those retrieved failed to be identified, potentially contributing to sustained horizontal transmission of *C. auris* outside the ICU setting. Previous molecular epidemiological investigations of *C. auris* outbreak isolates from our institution revealed the epidemic was predominantly monoclonal [20,21], strengthening the hypothesis of a common, persistent source of *C. auris* spread in our endemic setting. Another hypothesis might be that nosocomial or community transmission of *C. auris* is ongoing outside the endemic ICUs. This last hypothesis seems less plausible since no secondary outbreaks were detected starting from these additional detected cases of *C. auris* infection or colonization, but this will need to be further confirmed with genomic studies.

With regards to invasive infections, 10% of the cases of first detection of *C. auris* presented with candidemia, and secondary development of candidemia in colonized patients occurred frequently [2]. However, the incidence of candidemia in our cohort might be underestimated since we only considered patients who continued to be followed at our center, even though the risk of candidemia among discharged colonized patients is expected to decrease over time.

Limitations of our study are its retrospective and single center nature. However, this study offered unique and original data on the management of *C. auris* in endemic and locally restricted settings. Moreover, as already mentioned above, no genotypic study was performed on all included samples, preventing a firm conclusion on the possible origin of acquisition of colonization, although genotypic data for a subgroup of patients from our institution fully confirmed a clonal outbreak [20,21].

In conclusion, in settings where *C. auris* transmission is limited to a specific setting (e.g., ICU), dedicated policies should be implemented to maximize the effort to limit the spread outside the endemic areas. Our findings support the need to timely screen patients after discharge from the endemic areas and suggest that repeated screening over time could be beneficial in order not to miss late-onset cases of *C. auris* colonization/infection.

## Figures and Tables

**Figure 1 jof-10-00026-f001:**
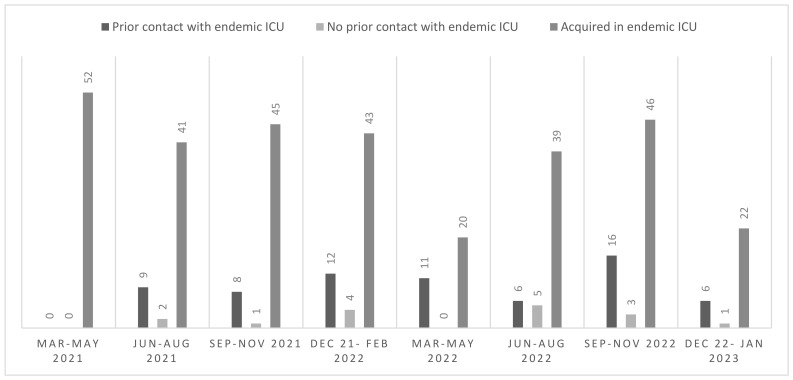
Epidemiological curve of new cases of *C. auris* colonization or infection in different periods, divided into those detected during the stay in endemic ICUs and those detected outside endemic ICUs, both with or without prior contact with endemic ICUs.

**Figure 2 jof-10-00026-f002:**
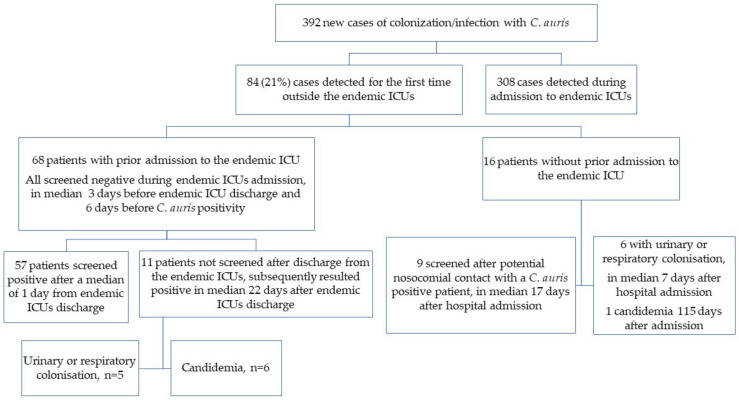
Flowchart of patients included in this study.

**Table 1 jof-10-00026-t001:** Demographic and microbiological characteristics of patients receiving the first diagnosis of colonization/infection with *C. auris* outside the endemic ICUs.

Patients’ Demographic Characteristics	Overall (*n* = 84, 100%)	Prior Endemic ICUs Contact (*n* = 68, 81%)	No Prior Endemic ICUs Contact (*n* = 16, 19%)
Male sex	56 (66.7%)	46 (67.6%)	10 (62.5%)
Median age, years (IQR)	70 (54–76)	66 (48–75)	75 (70–89)
Overall mortality	29 (20.6%)	19 (27.9%)	10 (62.5%)
**Microbiological results**			
Site and method of first detection of colonization or infection			
* Skin swab (all types of tests)*	64 (76.2%)	55 (80.9%) *	9 (56.3%)
*Molecular detection*	24/64 (37.5%)	18/55 (32.7%)	6/9 (66.7%)
*Conventional culture*	40/64 (62.5%)	37/55 (67.3%)	3/9 (33.3%)
*Respiratory*	6 (7.1%)	3 (4.4%)	3 (18.8%)
*Urinary*	6 (7.1%)	3 (4.4%)	3 (18.8%)
*Candidemia*	8 (9.5%)	7 (10.3%)	1 (6.3%)
Median time from ICU discharge to first detection of colonization/infection, days (IQR)	-	3 (1–8)	NA
Previous negative screening performed		68/68 (100%)	NA
*Molecular detection*	-	45 (66.2%)	NA
*Conventional culture*	-	23 (33.8%)	NA
Median time from last screening negative performed to first positivity (colonization or infection), days (IQR)	-	6 (4–12)	NA
Development of subsequent infection among 76 colonized patients			
*Candidemia*	3/76 (3.9%)	2 (3.0%) (after 23 and 31 days from colonization)	1 (6.3%) (after 19 days from colonization)
*Septic arthritis*	1/76 (1.3%)	1 (1.5%) (after 64 days from colonization)	0 (0.0%)

* Two additional patients were screened and tested positive in a skin swab performed after colonization of another site was already detected. NA: not applicable.

**Table 2 jof-10-00026-t002:** Proposed policy for the containment of *C. auris* spread in those institution where endemicity is limited to certain wards.

**Proposed Policy for the Containment of *C. auris* Outside an Endemic Setting**
**Sample for screening:** composite skin swab (axilla and groin, bilateral)
**Method for screening:** molecular-based tests for detection of *C. auris* DNA
**Timing of screening in the endemic setting**
1. Screening at admission to the endemic setting
2. Weekly screening throughout stay in the endemic setting, discontinued once the patient tests positive
3. Screening at the moment of discharge from the endemic setting if patient persistently negative
**Screening and management of *C. auris*-negative patients after transfer from an endemic setting**
1. Implement preemptive contact precautions
2. Perform screening at the moment of admission to a nonendemic ward and 48 h later
3. Discontinue preemptive contact precautions if both samples result negative
4. Repeat screening weekly to detected possible delayed-onset colonization
**Additional considerations**
The optimal length of screening remains unknown, but most of colonization cases were detected within 9 days from endemic ward discharge.
Prolonged screening, possibly weekly for the first 2 weeks of a hospital stay, might be needed to detect delayed-onset colonization.
Consider the risk of *C. auris* candidemia as the first clinical manifestation in patients not properly screened.

## Data Availability

Data are available from the corresponding author upon a request. The data are not publicly available due to privacy reasons.

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
