# Peer review of "Frequency of Detection of Candida auris Colonization Outside a Highly Endemic Setting: What Is the Optimal Strategy for Screening of Carriage?"

_jof, 2023, doi:10.3390/jof10010026_

Round 1

Reviewer 1 Report

Comments and Suggestions for Authors

This study proposed the described of the occurrence of new C. auris colonization/infection outside the endemic wards, in order to  add evidence for future policies on screening for colonization among negative patients   discharged from an endemic seing into other healthcare units. 

 However, the authors have some problems in text of methodologies from isolation C.auris strains for the study

The authors should  include this important adjusts in text that increase quality of the manuscript.

Below is included this important question about methodology:

The authors should describe the methodology applied for the isolation of Candida auris . What was medium used?

After include methodology, please review results.

Author Response

Dear Editor,

Please find enclosed a point-by-point response to the comments received from reviewers. Edited text is highlighted in red color. We added a few references that we hope might be included in the final text.

We hope that the amended version of the paper will be deemed suitable for publication in Journal of Fungi.

Best regards,

Prof. M. Mikulska, MD, PhD

Dr. L. Magnasco, MD

REVIEWER 1

C: “The authors should describe the methodology applied for the isolation of Candida auris. What was the medium used?”

A: As suggested, details about the cultural approach employed for the isolation of C. auris have been added to the text. (lines135-137)

Reviewer 2 Report

Comments and Suggestions for Authors

The manuscript of Magnasco and Mikulska et al. entitled: “Detection of Candida auris Colonization Outside a Highly Endemic Setting: What Is the Optimal Strategy for Screening of Carriage?” provides insights into the epidemiological situation and infection control of C. auris in an hospital setting, where C. auris is found to be endemic in 2 of 5 ICU units. Furthermore, it gives valuable information about the amendment of infection control procedures in this very unique setting. However, some important textual and formal aspects need to be addressed to further scientifically fortify this work:

Major

-Can you include the MIC data in the manuscript? These are mentioned in the methods section, but data is not shown or discussed. Did you observe phenotypic /genotypic echinocandin resistance?

-Can you include a section about the treatment of C. auris candidemia? How were the patients treated and did they survive?

- The result section should be written more concisely and shortened. The focus and text flow should be as follows:

-Data from patients without prior contact TO the endemic ICUs, please define contact! Data from patients with prior contact TO the endemic ICUs. Patients with candidemia (please shorten the section and include therapeutical data (see above)) New section with MIC values and a comparison of culture and PCR data?! Importantly, how many and which patients were positive in both methods? How many were positive in only one? Differences in infection / colonization?

-Please divide demographic from microbiological findings more prominently in table 1. Include mortality of infected / colonized patients.

-Figure2: All letters should have the same size and should not be squeezed in the boxes! In one box data is missing!! It says: “positive later in XX after endemic ICU discharge” Please amend.

-Please discuss your proposed algorithm of screening more prominently in the discussion section! How are your recommendations supported by your findings. Explain in the manuscript!

-Table 2: Why only molecular based tests? Sensitivity? Please explain.

-The manuscript should be revised through a native speaker solving grammatical issues.

Minor

- ll 38- 42: Can you include possible reasons why c. auris persists in the hospital?

-Lines 84-86: please erase. This is already stated in the introduction. Please include the secondary aim there.

Ll 139-140: Please rephrase the sentence. The objectives remain unclear.

Ll: 220-221: Among those not these. Please correct.

Comments on the Quality of English Language

grammar can be improved

Author Response

Dear Editor,

Please find enclosed a point-by-point response to the comments received from reviewers. Edited text is highlighted in red color. We added a few references that we hope might be included in the final text.

We hope that the amended version of the paper will be deemed suitable for publication in Journal of Fungi.

Best regards,

Prof. M. Mikulska, MD, PhD

Dr. L. Magnasco, MD

REVIEWER 2

C1: “Can you include the MIC data in the manuscript? These are mentioned in the methods section, but data is not shown or discussed. Did you observe phenotypic /genotypic echinocandin resistance?”

A1: We thank the Reviewer for the comment. In our hospital AFST is routinely performed on clinically significant isolates only, hence we cannot provide data on isolates cultured from screening samples. However, extensive phenotypic (MIC) and genotypic (WGS) data of isolates from the outbreak detected at our institution (including two cases of echinocandin resistance) have been previously published in dedicated studies (10.3390/jof7020140 and 10.2807/1560-7917.ES.2023.28.14.2300161). The methods section has been modified to clarify this point (line 141), and we added a dedicated paragraph about MIC values for invasive isolates, as requested below (lines 183-190), and provided the data from the aforementioned studies.

C2: “Can you include a section about the treatment of C. auris candidemia? How were the patients treated and did they survive?”

A2: We added requested data on treatment of invasive infections and the outcome of those patients in the specific paragraph about candidemia (lines 263-270)

C3: “The result section should be written more concisely and shortened.”

A3: We were invited by the Editor to increase the number of characters of the text of the paper in order to meet the requested length for publication in Journal of Fungi. We changed the order of the paragraphs as suggested by Reviewer (first paragraph about AFST, then about patients without any contact to the endemic ICUs). We hope that now the results seem easier to follow.

C4: “Data from patients without prior contact TO the endemic ICUs, please define contact!”

A4: Thank you for the comment, we added a definition of contact to the endemic ICUs in the specific Material and Methods paragraph (lines 98-99)

C5: “New section with MIC values and a comparison of culture and PCR data?! Importantly, how many and which patients were positive in both methods? How many were positive in only one? Differences in infection / colonization?”

A5: Thank you for the comment. We added the data on MIC values as requested above (lines 183-190). We also added information about the differences in sensitivity and turnaround time of culture and molecular tests in the results section (lines 133 and 321-336), as suggested by other Reviewer as well. Regarding the number of patients with both conventional culture and molecular screening positive samples, they were very few since we do not routinely perform culture in case of molecular positivity. Indeed, in clinical practice one test is used in place of the other, although for scientific purposes and internal validation of the test a certain amount of skin swabs collected for molecular testing were also processed for conventional culture from our Microbiology laboratory (unpublished data, not shown because not all the patients included in this study were processed). -The different number of patients with infection vs colonization is stated in the main text as well as in Table 1. It must be stressed that molecular tests are used only for screening in skin swabs, while all other samples are processed with conventional culture – we clarified this point in the dedicated paragraph of microbiological analysis (lines 124-130).

C6: “Please divide demographic from microbiological findings more prominently in table 1. Include mortality of infected / colonized patients.”

A6: Thank you for the suggestion. We added a division in Table 1 to better clarify these data. Mortality of infected/colonized patients is already shown in the fourth row of Table 1, as requested we added mention in the text to the mortality in the subgroup of patients with invasive infection (see above-lines 268-270)

C7: “Figure2: All letters should have the same size and should not be squeezed in the boxes! In one box data is missing!! It says: “positive later in XX after endemic ICU discharge” Please amend.”

A7: We are sorry for the mistake in submitting the paper. We corrected Figure 2 to match the format of the main text and amended missing data.

C8: Please discuss your proposed algorithm of screening more prominently in the discussion section! How are your recommendations supported by your findings. Explain in the manuscript!

A8: Thank you for your comment. We expanded the Discussion section accordingly (lines 329-333).

C9: “Table 2: Why only molecular based tests? Sensitivity? Please explain.”

A9: Thank you for the suggestion. Molecular screening was proposed since it provides a shorter time-to-result compared to conventional cultural workflow, generating actionable results and leading to a timely adoption of proper infection control strategies in the context of a nosocomial outbreak or the risk of potential horizontal transmission. The sensitivity of the molecular test used in this study was specified in the methods section (line 133). Moreover, we added an explanation on the reasons we decided to recommend molecular testing in the results section (lines 321-336).

C10: “The manuscript should be revised through a native speaker solving grammatical issues.”

A10: Thank you for your comment. We once more revised the text to amend possible grammar errors.

C11: “ll 38- 42: Can you include possible reasons why c. auris persists in the hospital?”

A11: We added a brief explanation of the reasons we suppose lead to the persistence of C. auris in our hospital in the Introduction section (lines 44-46)

C12: “Lines 84-86: please erase. This is already stated in the introduction. Please include the secondary aim there.”

A12: Thank you for your suggestion. We moved mention to the secondary aim also in the Introduction section (lines 71-72).

C13: “Ll 139-140: Please rephrase the sentence. The objectives remain unclear.”

A13: We are sorry but we could not address this revision as no objectives are stated in the lines cited and we could not understand the revision requested. Could you please report the sentence that you would like us to modify?

C14: “Ll: 220-221: Among those not these. Please correct.”

A13: Thank you for the suggestion, we decided to write “among them (i.e., the patients)”.

Reviewer 3 Report

Comments and Suggestions for Authors

Comments to the authors:

1. The importance of performing screening for Candida auris on patients transferred from endemic ICUs is well understandable. However, the reviewer feels uncertain whether using molecular methods for screening is cost-effective, even though the turnaround time is shorter than conventional method. Conventional culture method along with the identification by MALDI TOF-MS could be a reliable method with less costs. The authors should compare the merits and disadvantages of both molecular and conventional methods.

2. The authors proposed that patients who are transferred from an endemic ward to a non-endemic ward should be screened at the moment of transfer and 48 hours later. However, in Table 1, the authors showed that the median time from last negative screening to first positivity was 6 days (IQR, 4-12) in patients who were transferred from an endemic ward. From this result, it seems to be more reasonable to screen such patient at the moment of transfer and at least one week later, and until a negative result was confirmed contact precaution should be continued. Furthermore, additional weekly surveillance cultures might be considered since surveillance culture may become positive even after repeated negative results.

3. Patients who were admitted in an endemic ward and discharged may be colonized by Candida auris for a long term. The authors should propose a screening policy for re-admitted patients who has a history of admission in an endemic ward.

Author Response

Dear Editor,

Please find enclosed a point-by-point response to the comments received from reviewers. Edited text is highlighted in red color. We added a few references that we hope might be included in the final text.

We hope that the amended version of the paper will be deemed suitable for publication in Journal of Fungi.

Best regards,

Prof. M. Mikulska, MD, PhD

Dr. L. Magnasco, MD

REVIEWER 3

C1: “1. The importance of performing screening for Candida auris on patients transferred from endemic ICUs is well understandable. However, the reviewer feels uncertain whether using molecular methods for screening is cost-effective, even though the turnaround time is shorter than conventional method. Conventional culture method along with the identification by MALDI TOF-MS could be a reliable method with less costs. The authors should compare the merits and disadvantages of both molecular and conventional methods.”

A1: Thank you for your suggestion. As requested by another Reviewer as well, we added a few lines about the differences in sensitivity and turnaround time of culture and molecular tests in the discussion section (lines 321-336). As anticipated, cultural methods are more cost-effective than molecular ones and easy to implement, since they did not differ from conventional yeast cultures routinely performed in the clinical microbiology laboratories. However, it should be noted that they’re affected by several limitations. First, yeast growth is slow, and cultures may require up to 10 days to be deemed negative (10.1111/imj.14612); furthermore, the correct detection of C. auris can be challenging for the diagnostic laboratories, due to the need of updated instruments’ databases and ideally MALDI-TOF MS technology to obtain reliable specie-level identification. Even if adequate instruments are available, C. auris is hardly distinguishable from other Candida spp. on standard agar plates and, although specifically formulated chromogenic media or selective growth conditions may prove helpful, every suspicious colony should be identified to rule out the presence of C. auris, making the whole process time consuming. In case of abundant growth of other yeasts of similar appearance, few colonies of C. auris can be missed causing a delay in the detection of colonization and in subsequent cohorting. On the other hand, molecular methods are more costly, but retain a higher sensitivity and can grant more reliable and definitive results in a limited timeframe (hours). Considering that patients’ colonization can occur in just 4 hours from admission to a C. auris endemic environment (10.1186/s13756-016-0132-5), fast and accurate detection of colonization appear to be of paramount importance to help the correct management of infection control practices.

C2: “The authors proposed that patients who are transferred from an endemic ward to a non-endemic ward should be screened at the moment of transfer and 48 hours later. However, in Table 1, the authors showed that the median time from last negative screening to first positivity was 6 days (IQR, 4-12) in patients who were transferred from an endemic ward. From this result, it seems to be more reasonable to screen such patient at the moment of transfer and at least one week later, and until a negative result was confirmed contact precaution should be continued. Furthermore, additional weekly surveillance cultures might be considered since surveillance culture may become positive even after repeated negative results.”

A2: Thank you for your comment. As outlined in Table 2, the analysis of the data of the present study led us to recommend repeated screening of patients even after the first two screenings already suggested within 72h of discharge from the endemic ICUs. We better discussed this point, as suggested, in the Discussion section. Also, we further analyzed data (see Results section – lines 234-236) and found out that this quite long gap from last negative screening to first detection of colonization is dependent on a delay in screening (no patient had repeated negative screening before delayed detection of colonization, they simply were not screened anymore). At the present moment, we know that the timepoints we propose for screening are controversial, but our decision is dependent on the feasibility of maintaining contact precautions for long periods of time in a center caring for a high volume of patients at risk (hundreds of patients discharged from endemic ICUs each year) and on the balance between benefits for local infection control and feasibility of the measures proposed. It might differ in setting with a very limited number of cases.

C3: “Patients who were admitted in an endemic ward and discharged may be colonized by Candida auris for a long term. The authors should propose a screening policy for re-admitted patients who has a history of admission in an endemic ward.”

A3: Thank you very much for the comment. The internal protocol in use at our hospital does advocate for repeated screening for C. auris colonization for those patients with certain risk factors (including admission to the high endemicity ICUs in the last year) in the event of re-admission to any ward of our hospital. Moreover, the issue of duration of C. auris colonization is of uttermost importance to guide the long-term management of C. auris-colonized patients. Even though these topics are not pertinent with the main focus of the present work, we mentioned this issue in the paragraph about the proposed algorithm for screening (lines 281-284).

Reviewer 4 Report

Comments and Suggestions for Authors

• The manuscript presented is undoubtedly of utmost relevance due to the detection of one of the main species of Candida involved in recent years as a causal agent of in-hospital outbreaks in critically ill patients. And in the current pandemic it is necessary to have information about it.

• I would like to make the following observations that could improve the quality of the manuscript.

• It is advisable that the title of the manuscript be changed, since it does not agree with the objective presented in the abstract as well as in the results presented. The title should be focused on: Epidemiology of…., Frequency of isolation of…. Etc.

• It is strange that cases of lung colonization/infection have been reported. The criterion for colonization by candida is through biopsies. Did they do it?

• Section 2.1: The wording of this and the entire document must be written in the past verb.

• Section 2.1 “The main objective of this study was to describe the epidemiology of C. auris colonization/infection cases detected outside the two endemic ICUs. The secondary aim was to 85 propose an algorithm for testing of patients at risk outside the highly endemic setting.” It is not described in the abstract, and furthermore all the objectives are worded differently in the abstract and this section.

• It is convenient that for each type of infectious event, the symptoms of candida infection are indicated.

• In my experience, the extraction of DNA from yeast is complex due to the chemical nature of the cell wall. The authors indicate that they used a DNA extraction kit for whole blood. It's right? Didn't they couple their DNA extraction with a specific enzyme such as lyticase, zymolase? The kit used alone is not capable of lysing fungal cells. Review and clarify.

• In the document they refer to horizontal transmission, assuming possible contamination between patients. Can you verify this? Molecular methods for the detection of clones in yeast could be useful, such as the T3B method, RAP-PCR, among others. I consider they could implement some molecular test to be able to affirm that there is contamination or horizontal transmission. If it is not within your reach, it is advisable to describe a section of the study's weaknesses, where the lack of molecular tests is exposed. Currently, classical epidemiology is no longer sufficient to establish the triad: patient, environment and infectious agent.

• Section 3. Results. "3. During the study period, we overall observed 392 incident cases of colonization or infection with C. auris in our hospital. Among these, 84 (21.4%) were detected outside the endemic ICUs.” It is worrying that “they do not differentiate between colonization and infection.”

• Figure 1. It is an epidemic curve of cases, it is convenient that you use this term during the manuscript.

• Table 1. “Molecular” change to “molecular detection”

Author Response

Dear Editor,

Please find enclosed a point-by-point response to the comments received from reviewers. Edited text is highlighted in red color. We added a few references that we hope might be included in the final text.

We hope that the amended version of the paper will be deemed suitable for publication in Journal of Fungi.

Best regards,

Prof. M. Mikulska, MD, PhD

Dr. L. Magnasco, MD

REVIEWER 4

C1: “It is advisable that the title of the manuscript be changed, since it does not agree with the objective presented in the abstract as well as in the results presented. The title should be focused on: Epidemiology of…., Frequency of isolation of…. Etc.”

A1: Thank you for the suggestion. We modified the title as suggested.

C2: “It is strange that cases of lung colonization/infection have been reported. The criterion for colonization by candida is through biopsies. Did they do it?”

A2: Thank you for the comment. No respiratory invasive infection is reported, but only candidemias and an episode of septic arthritis, and all the cases of isolation of C. auris from respiratory specimens were considered as colonizations, as stated in the Methods section and in Table 1.

C3: “Section 2.1: The wording of this and the entire document must be written in the past verb.”

A3: Thank you for the comment, we amended accordingly. We maintained present tense only in the paragraph about the proposed algorithm for screening.

C4: “Section 2.1 “The main objective of this study was to describe the epidemiology of C. auris colonization/infection cases detected outside the two endemic ICUs. The secondary aim was to 85 propose an algorithm for testing of patients at risk outside the highly endemic setting.” It is not described in the abstract, and furthermore all the objectives are worded differently in the abstract and this section.”

A4: Thank you, we added mention to the secondary aim also in the abstract (line 19).

C5: “It is convenient that for each type of infectious event, the symptoms of candida infection are indicated.”

A5: Thank you for your comment. We decided not to include data on symptoms of invasive C. auris infections (which were in all cases but one candidemias) since they are not in the focus of the topic treated in the present study, but most of patients presented with fever. Local symptoms and fever were present in the case of arthritis (lines 106-108). We added in the text that sampling for invasive infections was performed in presence of symptoms of systemic infection in the Methods section.

C6: “In my experience, the extraction of DNA from yeast is complex due to the chemical nature of the cell wall. The authors indicate that they used a DNA extraction kit for whole blood. It's right? Didn't they couple their DNA extraction with a specific enzyme such as lyticase, zymolase? The kit used alone is not capable of lysing fungal cells. Review and clarify.”

A6: Thank you for the comment. We agree with the Reviewer’s comment, in that DNA purification from yeast is usually more technically demanding than that from bacteria. The DNA purification kit used in the study (validated for use with the RealCycler® CANDIDA AURIS PCR KIT) is designed for total DNA purification from whole blood, combining an initial heat-based lysis and then proteinase K and guanidine hydrochloride for cell lysis and protein degradation. A comment has been added to the text to clarify this point (lines 124-130).

C7: “In the document they refer to horizontal transmission, assuming possible contamination between patients. Can you verify this? Molecular methods for the detection of clones in yeast could be useful, such as the T3B method, RAP-PCR, among others. I consider they could implement some molecular test to be able to affirm that there is contamination or horizontal transmission. If it is not within your reach, it is advisable to describe a section of the study's weaknesses, where the lack of molecular tests is exposed. Currently, classical epidemiology is no longer sufficient to establish the triad: patient, environment and infectious agent.”

A7: Thank you for your comment. We did not perform genotypic studies on all the samples included, so we mentioned this among weakness in the paragraph about limitations of the study (lines 388-390). Comprehensive molecular-epidemiological investigations based on whole-genome sequencing of outbreak isolates (including two echinocandin resistance cases) have been previously published in dedicated studies (10.2807/1560-7917.ES.2023.28.14.2300161and 10.3390/jof7020140), demonstrating the monoclonal nature of the epidemic. As such, horizontal transmissions were deemed as primary cause of C. auris persistence in our setting. A comment has been added to the text to clarify this point.

C8: “Section 3. Results. "3. During the study period, we overall observed 392 incident cases of colonization or infection with C. auris in our hospital. Among these, 84 (21.4%) were detected outside the endemic ICUs.” It is worrying that “they do not differentiate between colonization and infection.”

A8: Thank you for your comment. For epidemiological purposes, only aggregated data are reported here, and we specified this in the text. All first episodes of detection of C. auris colonization are included for the scope of the study (whether blood samples or screening samples from non-sterile sites). However, separate data on colonization and infection are provided for the patients included in the study (lines 154-156)

C9: “Figure 1. It is an epidemic curve of cases, it is convenient that you use this term during the manuscript.”

A9: Thank you for the suggestion, we corrected the text accordingly.

C10: “Table 1. “Molecular” change to “molecular detection”

A10: Thank you for the suggestion, we corrected the text accordingly.

Round 2

Reviewer 3 Report

Comments and Suggestions for Authors

The concerns raised by the reviewer has been resolved.

Reviewer 4 Report

Comments and Suggestions for Authors

The authors have resolved the observations satisfactorily. I recommend your acceptance